# Predicting Beta-Lactam Target Non-Attainment in ICU Patients at Treatment Initiation: Development and External Validation of Three Novel (Machine Learning) Models

**DOI:** 10.3390/antibiotics12121674

**Published:** 2023-11-28

**Authors:** André Wieringa, Tim M. J. Ewoldt, Ravish N. Gangapersad, Matthias Gijsen, Nestor Parolya, Chantal J. A. R. Kats, Isabel Spriet, Henrik Endeman, Jasper J. Haringman, Reinier M. van Hest, Birgit C. P. Koch, Alan Abdulla

**Affiliations:** 1Department of Hospital Pharmacy, Erasmus University Medical Center, Dr. Molewaterplein 40, 3015 GD Rotterdam, The Netherlands; tim.ewoldt@gmail.com (T.M.J.E.); r.gangapersad@erasmusmc.nl (R.N.G.); b.koch@erasmusmc.nl (B.C.P.K.); a.abdulla@erasmusmc.nl (A.A.); 2Rotterdam Clinical Pharmacometrics Group, Erasmus University Medical Center, Dr. Molewaterplein 40, 3015 GD Rotterdam, The Netherlands; 3Department of Clinical Pharmacy, Isala Hospital, Dr. van Heesweg 2, 8025 AB Zwolle, The Netherlands; 4Department of Intensive Care, Erasmus University Medical Center, 3015 GD Rotterdam, The Netherlands; h.endeman@erasmusmc.nl; 5Clinical Pharmacology and Pharmacotherapy, Department of Pharmaceutical and Pharmacological Sciences, KU Leuven, 3000 Leuven, Belgium; matthias.gijsen@uzleuven.be (M.G.); isabel.spriet@uzleuven.be (I.S.); 6Pharmacy Department, UZ Leuven, 3000 Leuven, Belgium; 7Delft Institute of Applied Mathematics, Mekelweg 4, 2628 CD Delft, The Netherlands; n.parolya@tudelft.nl; 8Department of Hospital Pharmacy, Haaglanden Medical Center, Lijnbaan 32, 2512 VA The Hague, The Netherlands; c.kats@haaglandenmc.nl; 9Department of Intensive Care, Isala Hospital, Dr. van Heesweg 2, 8025 AB Zwolle, The Netherlands; j.haringman@isala.nl; 10Department of Pharmacy and Clinical Pharmacology, Amsterdam UMC, University of Amsterdam, Meibergdreef 9, 1105 AZ Amsterdam, The Netherlands; r.m.vanhest@amsterdamumc.nl

**Keywords:** target attainment, risk model, beta-lactams, penicillin, cephalosporin, carbapenem, critical illness, intensive care unit

## Abstract

In the intensive care unit (ICU), infection-related mortality is high. Although adequate antibiotic treatment is essential in infections, beta-lactam target non-attainment occurs in up to 45% of ICU patients, which is associated with a lower likelihood of clinical success. To optimize antibiotic treatment, we aimed to develop beta-lactam target non-attainment prediction models in ICU patients. Patients from two multicenter studies were included, with intravenous intermittent beta-lactam antibiotics administered and blood samples drawn within 12–36 h after antibiotic initiation. Beta-lactam target non-attainment models were developed and validated using random forest (RF), logistic regression (LR), and naïve Bayes (NB) models from 376 patients. External validation was performed on 150 ICU patients. We assessed performance by measuring discrimination, calibration, and net benefit at the default threshold probability of 0.20. Age, sex, serum creatinine, and type of beta-lactam antibiotic were found to be predictive of beta-lactam target non-attainment. In the external validation, the RF, LR, and NB models confirmed good discrimination with an area under the curve of 0.79 [95% CI 0.72–0.86], 0.80 [95% CI 0.73–0.87], and 0.75 [95% CI 0.67–0.82], respectively, and net benefit in the RF and LR models. We developed prediction models for beta-lactam target non-attainment within 12–36 h after antibiotic initiation in ICU patients. These online-accessible models use readily available patient variables and help optimize antibiotic treatment. The RF and LR models showed the best performance among the three models tested.

## 1. Introduction

### Background

The use of antibiotics in intensive care units (ICUs) is ten times higher than in other wards [1]. Despite this high use of antibiotics, infection-related mortality remains high in ICU patients at 30% [1,2]. To adequately prevent and treat severe infections in critically ill patients, it is important that patients are treated with an appropriate dosing regimen of antibiotics [2]. However, the dosing regimens used in ICU patients are designed for non-severely ill patients or derived from studies in healthy volunteers and might result in suboptimal antibiotic exposure [3]. Furthermore, ICU patients are a highly heterogenic group of patients that undergo extensive physiological alterations (i.e., renal or hepatic dysfunction, altered fluid status, changes in albumin concentration), which have a potential impact on antibiotic pharmacokinetics (PK) and thereby the exposure of antibiotics [2,3,4,5].

Achievement of unbound plasma concentrations above the minimal inhibitory concentration (MIC) for a certain fraction of the dosing interval (*f*T > MIC) is associated with a higher likelihood of clinical cure and bacteriological eradication with a decrease in the potential for antimicrobial resistance against beta-lactam antibiotics [6,7,8,9,10,11]. Different pharmacodynamic targets (PDTs) have been identified, from one up to four times the MIC for 100% of the dosing interval (100%ƒT > 1–4 × MIC) [6,7,8,9,10]. Currently, achieving the 100% *f*T > MIC target is recommended for beta-lactam antibiotics in ICU patients [2]. However, a significant percentage of ICU patients (37–45%) fail to achieve this PDT [2,6,10].

Adequate target attainment can be anticipated in ICU patients before initiation of beta-lactam antibiotic therapy based on demographic and clinical factors such as sex, age, and renal clearance [12]. In addition, the routine use of therapeutic drug monitoring (TDM) for beta-lactam antibiotics has the potential to maximize therapeutic success and is therefore recommended in ICU patients [2,6,7].

To maximize the efficient use of limited hospital resources, TDM should ideally be performed only in ICU patients at risk of target non-attainment or toxicity. A prediction model could aid in identifying patients who are at risk of target non-attainment or toxicity and would benefit from TDM. To date, there is still a lack of prediction models for beta-lactam target non-attainment in the ICU setting.

Therefore, this study aimed to develop and externally validate broadly applicable diagnostic multivariable prediction models that are able to predict the probability of target non-attainment in ICU patients at the initiation of beta-lactam therapy.

## 2. Results

### 2.1. Model Building and Internal Validation

#### 2.1.1. Model Development

A total of 376 patients were included in the development and internal validation of the three models (see Appendix A). The descriptive statistics of the variables of interest are shown in Table 1. More descriptive statistics of the variables of interest between patients reaching the PDT or not are shown in Appendix B. The prescribed dosages of the antibiotics are available in Appendix C. The predefined PDT is a trough level (C_trough_) > MIC epidemiologic cut-off value (MIC_ECOFF_) and this was not attained in 115 patients (30.6%), as shown in Table 1.

The Boruta algorithm identified the variables age, sex, serum creatinine, and type of antibiotic. No other relevant variables were found after the manual addition and removal of potential variables. Random forest (RF), logistic regression (LR), and naïve Bayes (NB) models were built using these four variables. For the RF model, the relative importance of the four variables was calculated, and serum creatinine and the type of antibiotic were found to have the largest contribution to target non-attainment (see Appendix D).

The LR model was used to calculate the odds ratios of the different variables (Table 2). Among the beta-lactam antibiotics, patients who received cefotaxime had the highest likelihood of not attaining the target (Table 2).

#### 2.1.2. Internal Validation

The ideal threshold probability for the RF, LR, and NB models was 0.20, 0.21, and 0.17, respectively, with comparable results with the 0.20 threshold probability. The NB, LR, and RF models showed an area under the receiver operating characteristic (AUROC) curve of 0.82 [95% CI 0.74–0.91], 0.81 [95% CI 0.76–0.91], and 0.81 [95% CI 0.75–0.91], respectively (Figure 1A). For the specific values with 95% CI of all metrics with the ideal and 0.20 threshold, see Appendix E.

Decision curve analysis (DCA) showed that the three models presented had a net benefit for the RF, LR, and NB models over the “Treat all patients” and “Treat none” over a broad range of threshold probabilities targeted: 0.06–0.60, 0.10–0.68, and 0.10–0.59, respectively (Figure 2A).

Classifying a patient as showing target non-attainment within 12–36 h will trigger the physician to consider therapeutic drug monitoring. Accordingly, lower or higher threshold probabilities are used when harm associated with a positive prediction is rather low or high, respectively. Considering antimicrobial therapy, the harm associated with a positive prediction for beta-lactam target non-attainment within 12–36 h is relatively low; hence, lower threshold probabilities are adequate. The threshold probability is an interpretation of the harm-to-benefit ratio between false positives and true positives. A prediction model is clinically useful if its net benefit is higher than for the alternative strategies (i.e., the line of the model has to be above the “Treat all patients” and “Treat none” lines at the chosen threshold probability) [13]. As shown in this figure, our target non-attainment models show net benefit above the alternative strategies over a broad range of threshold probabilities both in the internal validation set (all models) and external validation set (the LR and RF models) and, as a result, show clinical utility in those ranges. The ideal threshold probability for the models was ~0.20, which is an adequate threshold for many clinical decisions concerning antimicrobial dosing in the ICU. Depending on the clinical context, the threshold probability can be shifted upwards or downwards.

The calibration plots demonstrate excellent calibration of the RF and NB models over a broad range of threshold probabilities with a slope of 1, intercept of 0.0 for both models, and an expected calibration error (ECE) of 0.02 and 0.09, respectively. The LR model also shows good calibration with a slope of 1.1, intercept of 0.0, and an ECE of 0.06 (see Appendix F).

### 2.2. External Validation

A validation cohort of 150 patients was used after excluding 3 patients undergoing renal replacement therapy (RRT). The target was not attained in 72 patients (48.0%). For more information on the descriptive statistics of the variables of interest, refer to Appendix G. The prescribed dosages of the antibiotics are shown in Appendix H.

The median serum creatinine concentration was significantly lower in the external validation set compared with the internal validation set, at 0.77 mg/dL [0.57–1.04] and 0.995 mg/dL [0.69–1.56] (*p* < 0.001), respectively. Significant differences between the type of antibiotic used were also found between the validation datasets (*p* < 0.001) (see Appendix I).

DCA showed that the RF, LR, and NB models had a net benefit over the “Treat all patients” and “Treat none” strategies over a wide range of threshold probabilities: 0.06–0.80, 0.15–0.93, and 0.38–0.88, respectively (see Figure 2B). However, the NB model showed a negative net benefit compared with “Treat all patients” below the threshold probability of 0.38.

At the 0.20 threshold probability, the RF, LR, and NB models showed an AUROC of 0.75 [95% CI 0.67–0.82], 0.80 [95% CI 0.73–0.87], and 0.79 [95% CI 0.72–0.86], respectively (Figure 1B). For the specific values with 95% CI of all metrics with the ideal and 0.20 threshold, see Appendix E.

The calibration plots showed that the RF, LR, and NB models had acceptable calibration over a wide range of threshold probabilities, with a slope of 1.3, 1.1, and 0.8, respectively, and an intercept of 0.0 in all models. For the RF, LR, and NB models the ECE was 0.12, 0.13, and 0.15 respectively (for a detailed description, Appendix F).

### 2.3. Online Beta-Lactam Target Non-Attainment Predictor

The online beta-lactam target non-attainment predictor is available at: https://cator010.nl/betalactampredictor (accessed on 26 September 2023).

## 3. Discussion

This study aimed to develop and externally validate multivariable prediction models to identify critically ill patients at risk for beta-lactam target non-attainment within 12–36 h of beta-lactam therapy. To our knowledge, these models are the first of their kind. Using these models, we developed a broadly applicable online dose-optimization tool. This tool can be used by physicians and pharmacists to maximize the potential for therapeutic success with beta-lactam therapy in ICU patients.

In terms of overall performance for the internal and external validation, the RF and LR models both show good discrimination, calibration, and net benefit for predicting of beta-lactam target non-attainment. The NB model demonstrated lower performance compared with the RF and LR models in terms of negative net benefit for predicting beta-lactam target non-attainment in the external validation cohort. As a result, the NB model is less suitable for this purpose.

We found that serum creatinine had the largest contribution to beta-lactam target non-attainment, followed by the type of antibiotic used, age, and sex (male); these findings are in line with the conclusions of Abdulla et al. [12]. Other potential risk factors have been proposed, but all showed non-significant impact in the current study, such as the risk factor lower daily dose, which is likely to be due to low variations in prescribed dosages. Furthermore, the risk factor high BMI has been proposed, potentially resulting in a larger volume of distribution and enhanced renal function due to increased kidney size and renal blood flow [12,14]. Finally, a low SOFA score has also been suggested, but this factor is correlated with renal function, which is already accounted for in the models [12].

In our cohort, ceftriaxone was the most prescribed antibiotic and was used as the reference beta-lactam antibiotic for the variable type of antibiotic. Compared with ceftriaxone, most of the other antibiotics had a higher probability of target non-attainment. The difference in selected MIC_ECOFF_ values for the beta-lactam antibiotics is one of the factors having an impact on attaining beta-lactam target non-attainment. All beta-lactam antibiotics undergo renal elimination, but other factors such as changes in protein binding and volume of distribution can also play a role in target non-attainment, especially for antibiotics with high plasma protein binding such as ceftriaxone and flucloxacillin.

With clinical chemistry data used from multiple hospitals, there is a risk of differences in reported concentrations of chemistry analytes. However, the risk is minimal because commonly used chemistry analyzers show similar concentrations when comparing chemistry analytes, such as serum creatinine, urea, and albumin [15].

Our beta-lactam target non-attainment models are readily available online. By default, these models have a threshold probability of 0.20, which is appropriate for many clinical decisions regarding antimicrobial dosing in the ICU. Moreover, this threshold closely matches the ideal threshold for the best performance metrics of both the RF and LR models. In our models, the clinician can choose a lower threshold probability when critically ill patients are on the ward. The threshold reflects the trade-off between harm associated with a false positive and benefit associated with a true positive outcome. Hence, the threshold of 0.20 reflects that benefit is four times greater than harm ([1–0.20]/0.20). When the calculated probability for beta-lactam target non-attainment within the first 12–36 h is higher than the threshold probability, the models will flag this patient as being at risk for target non-attainment. This may be a trigger for the clinician to consider beta-lactam TDM.

The manual selection of a threshold within our online beta-lactam target non-attainment models enables users to consider other threshold probabilities. The decision to choose a different threshold probability within the clinical context might be challenging for the clinician what is the most suitable. It is essential to choose a threshold that provides a net benefit. To assist clinicians in making an informed decision, Figure 1 and Appendix J provide the rationale behind selecting an appropriate threshold. In short, our beta-lactam target non-attainment models show net benefit above the alternative strategies over a broad range of threshold probabilities in both validation sets for the RF and LR models (Figure 1), confirming their clinical usefulness when used within this range.

When evaluating the three models tested for beta-lactam target non-attainment, the NB algorithm operates under the assumption that the variables are independent risk factors contributing to the same outcome value. However, this assumption does not hold for our model since certain variables, such as age and renally eliminated antibiotics, are correlated with renal function. This may explain why the NB model performs less favorably than the other models [16].

Our study did not demonstrate a performance advantage of RF over LR in developing clinical prediction models, which is in line with the conclusions of Christadoulou et al. [17]. The RF algorithm consists of many decision trees and has the advantage that any interaction or correlation between variables does not adversely affect classification. The LR algorithm can be prone to overfitting and can negatively impact the diagnostic accuracy of the LR algorithm in the general population [18].

This study has several strengths. First, we included patients from a variety of academic and peripheral hospitals, originating from high-quality prospectively collected databases. Second, we studied the performance of three different models in predicting beta-lactam target non-attainment in a heterogenous group of critically ill patients. This was carried out in the internal and external validation sets, allowing selection of the best-performing models suitable for usage in a broad ICU population and therefore increasing its generalizability. Third, the four variables required as input for the models are readily available to the clinician during the ICU stay, especially at ICU admission. Fourth, the models were validated within 12–36 h after initiation of beta-lactam therapy, allowing quick monitoring of patients at risk of target non-attainment. Lastly, the models are accessible online, making them easily applicable in daily clinical practice in the ICU. This can provide valuable support for clinicians and pharmacists in improving the treatment of severe infections in critically ill patients.

This study also has some limitations. First, it should be noted that in the external validation a selection bias may have occurred, because only the patients were included that met the inclusion criteria as used in the internal validation set. The external validation cohort differed somewhat from the internal validation cohort of patients, where the overall renal function was also significantly higher in the external validation cohort, mainly due to one study by Gijsen et al. including patients with sepsis with preserved or increased renal function [19]. This probably explains the higher rate of target non-attainment in the external validation set (i.e., 48 versus 31%). Consequently, the overall performance dropped slightly in the external validation set. However, the RF and LR models still performed well, demonstrating the robustness and generalizability of these prediction models. Second, not all the beta-lactam antibiotics were present in the external validation set or in the same proportions compared with the internal validation set. There were no patients with cefotaxime, cefuroxime, or flucloxacillin included in the external validation set. Additionally, the proportion of patients receiving meropenem or piperacillin–tazobactam was larger in the external validation set and some patients had a higher dose regimen of amoxicillin–clavulanic acid. Third, the models have been specifically developed and validated to accurately predict exposure using intermittent dose regimens and within the range of doses that have been investigated. Consequently, using the models outside the stated intermittent dose ranges or for dose regimens involving continuous or extended beta-lactam infusion may lead to inaccurate predictions. Nevertheless, the models can be used to identify patients who may be at risk of underexposure at the start of treatment, regardless of the infusion method used.

Fourth, to optimize treatment for worst-case infections, the highest MIC_ECOFF_ values of presumed pathogens per beta-lactam antibiotic were used to calculate target attainment. Although the MIC_ECOFF_ is in many situations similar to the clinical breakpoint, this approach may not be suitable for patients infected with pathogens that have higher susceptibility and require lower antibiotic concentrations. However, since the actual MIC of the microorganism is often unknown in clinical practice, using MIC_ECOFF_ values is justifiable to initiate treatment. Furthermore, it is important to note that MIC_ECOFF_ values can change over time. For example, recently (March 2023), the MIC_ECOFF_ for ceftriaxone susceptibility of *Enterobacteriaceae* was lowered from 1 mg/L to 0.125 mg/L, a change which could affect beta-lactam target non-attainment predictions. However, we regularly monitor the MIC_ECOFF_ values and adjust our models as needed to ensure the best possible prediction for beta-lactam target non-attainment.

Fifth, beta-lactam antibiotics have a short half-life of 1–2 h for all studied antibiotics in patients with normal renal function, except for ceftazidime and ceftriaxone with half-lives of 4–6 h and 8–10 h, respectively [20]. Because renal function has the largest contribution to target non-attainment, patients with normal or better renal function are at risk for beta-lactam target non-attainment. In those patients, steady-state concentrations after three–five half-lives are presumed within 12 h after the start of beta-lactam antibiotics with a short half-life, but this was not investigated. For the beta-lactam antibiotics ceftriaxone and to a lesser extent ceftazidime, steady-state concentrations are not reached when measuring early in the 12–36 h window. Therefore, there is a risk that dosages of ceftriaxone and ceftazidime will unnecessarily be raised with a greater risk of side effects that may occur. In our opinion, plasma concentrations of antibiotics need to be above the target MIC_ECOFF_ as soon as possible after the start of the antibiotic for optimal treatment in critically ill patients and this outweighs the small risk of side effects of antibiotics. Measuring a second plasma concentration is considered in those patients when steady-state concentrations are reached.

Sixth, we measured total drug concentrations with correction for protein binding based on the literature. Measuring unbound concentrations is desirable in critically ill patients since the ratio of bound and unbound drugs can be subjected to changes because of the disease characteristics of these patients. Seventh, serum creatinine was selected as a marker for renal function, but this may not reflect the actual renal function in a critically ill patient. Rapid changes in renal function may occur and a lag time in changes of serum creatinine values exist, resulting in under- or overestimating the actual renal function and leading to inadequate predictions from our models. Finally, the models have not yet been validated in a prospective clinical trial to confirm their ability to reliably identify patients with beta-lactam target non-attainment. However, we are planning to conduct such research in the near future.

## 4. Patients and Methods

Appropriate reporting of our beta-lactam target non-attainment models was performed according to the guidelines for the transparent reporting of a multivariable prediction model for individual prognosis or diagnosis (TRIPOD) [21].

### 4.1. Model Building and Internal Validation

#### 4.1.1. Development Cohort

Patient data were assembled from two prospective trials, which were conducted in a total of 11 academic and peripheral hospitals between 2016 and 2021 [3,10]. All patients who met the following criteria were included in the model-building phase to maximize the power and generalizability of the results: (1) admitted to the ICU, (2) aged ≥ 18 years, (3) treated with intravenous intermittent beta-lactam antibiotic therapy, and (4) availability of a C_trough_ within 12–36 h of starting antibiotic treatment with no dose adjustment or cessation of therapy during this time. The antibiotics included in the study were cefotaxime, ceftazidime, ceftriaxone, cefuroxime, amoxicillin, amoxicillin–clavulanic acid, flucloxacillin, piperacillin–tazobactam, and meropenem. Patients were excluded in case of pregnancy, receiving studied antibiotics only as prophylaxis within the context of selective digestive tract decontamination, or when admitted to medium care. Burnwound patients as well as patients receiving RRT were also excluded, because of the significantly altered PK of beta-lactam antibiotics.

#### 4.1.2. Pharmacodynamic Target Attainment

The ECOFF of the presumed pathogens, as defined by the European Committee on Antimicrobial Susceptibility Testing, was used for each of the study antibiotics (see Appendix K) [22]. Beta-lactam pharmacodynamic target attainment is defined as 100%ƒT > MIC_ECOFF_. The C_trough_ concentration was directly compared against the highest MIC_ECOFF_ of the presumed pathogens to determine whether target attainment was achieved. The unbound concentration of the beta-lactam antibiotics was calculated with correction for protein binding based on the literature [3,10]. Plasma concentrations were determined by multi-analyte liquid chromatography with tandem mass spectrometry (LC-MS-MS) methods in accordance with quality standards as described in the published articles [23,24,25,26].

#### 4.1.3. Variable Selection

The beta-lactam target non-attainment models were developed using variables selected from the current literature [12], expert consensus, and data availability. Given that we aimed to develop broadly applicable prediction models, all variables needed to be available in the patient files or able to be calculated at the initiation of antibiotic therapy. The variables included for testing were: (i) at ICU admission: age, height, body weight, sex, body mass index (BMI), Acute Physiology and Chronic Health Evaluation IV score, admission for cardiac surgery, admission for trauma, type of antibiotic, frequency of dosing, dose of antibiotic, dose of antibiotic divided by the defined daily dosage (normalized dosing), the number of days in the ICU before start of the antibiotic; and (ii) on the day of the start of the antibiotic therapy: Sequential Organ Failure Assessment score (SOFA), fluid balance, white blood cell count, C-reactive protein, serum creatinine, urea, and albumin. A detailed description of the tested variables can be found in Appendix L.

#### 4.1.4. Model Development

We used three different statistical methods, RF, LR, and NB, to analyze beta-lactam target non-attainment. These methods use different mathematical and computational techniques to learn patterns and make predictions or classifications, but differ in key ways. RF is a machine learning algorithm that combines the outputs of multiple decision trees, while LR is a statistical model that models the probability of a binary outcome. Naïve Bayes is a probabilistic model based on Bayes’ theorem, which assumes independence between the risk factors.

We randomly divided the development cohort into a training set (80%) and an internal validation set (20%). Patients were stratified evenly between both sets for beta-lactam target attainment and type of antibiotic. In the RF model, weights were added to correct for the imbalance in the percentage of patients attaining the target. In the RF model’s training set, we used the Boruta algorithm for variable selection, which is highly sensitive (nearly 100%) and selective in selecting relevant variables for model building [27,28,29]. To ensure a robust model, we ran the RF model with 2500 decision trees for each run [29,30]. For the LR and NB models, the same predictor variables were used as those resulting from the Boruta algorithm. The relative contribution of the variables (variable importance) to target non-attainment was determined with the RF model by calculating the average Gini index decrease with the 2500 trees used.

#### 4.1.5. Internal Validation

The performance of the models was subsequently assessed on the model building set using 1000-times repeated 5-fold cross-validation [31].

The AUROC curve was used. AUROC curves were calculated for each cross-validation to evaluate discrimination. The 95% CI for the AUROC curves was calculated using DeLong’s test [32]. In the ICU setting, an adequate threshold probability for many clinical decisions concerning antimicrobial dosing is 0.20 [33,34]. This means that TDM should be applied if there is a 20% or higher chance that the target is not achieved, in which case the clinician should be willing to perform TDM in four patients who do not actually show beta-lactam target non-attainment in the next 12–36 h (false positives) to treat one patient who truly shows beta-lactam target non-attainment in the next 12–36 h (true positive). For more information regarding the threshold probability, see Appendix J. We manually selected the ideal threshold probability for each prediction model, based on optimal sensitivity and specificity, using Youden’s J statistic [35]. We calculated sensitivity, specificity, negative predictive value, positive predictive value, and misclassification for beta-lactam target non-attainment within the next 12–36 h after therapy initiation, both for the 0.20 and ideal threshold probability. For our beta-lactam target non-attainment models, we prioritized sensitivity over specificity, to avoid missing patients with target non-attainment.DCA was performed to evaluate the net benefit [36,37,38].Platt scaling was used to calibrate the RF model and NB model [39]. Calibration was assessed using calibration plots for all three models (intercept, calibration slope, and ECE) [13].

### 4.2. External Validation

Patient data were assembled from five observational ICU studies conducted between 2013 and 2018: one study is unpublished (Ethics Committee Research UZ/KU Leuven, S58397) and four have been published [19,24,25,26]. The same inclusion and exclusion criteria were applied as described above for the development cohort. Model performance was assessed as described above for the internal validation set at the same threshold probabilities.

### 4.3. Statistical Analysis

All statistical analyses were performed in R (version 4.1.1) [40]. For the stratification, the R package splitstackshape (version 1.4.8) was used. For performing RF, LR, and NB, the packages h2o (version 3.36.0.4), caret (version 6.0-92), and naivebayes (version 0.9.7) were used, respectively. Discrimination of the prediction models was considered adequate with mean target values for AUROC, sensitivity, and specificity of ≥0.75, ≥0.80, and ≥0.60, respectively. The ideal threshold probability for each prediction model was manually selected to achieve an optimal combination of mean sensitivity ≥0.80 and specificity ≥0.60.

Pairwise deletion was performed when data were missing. For patient characteristics, continuous data were presented as median and interquartile range and categorical data were presented as count and percentage. The Wilcoxon sum rank test was used for continuous variables, the chi-squared test for the categorical variables with categories with more than five observations, and the Fisher exact test for categories with fewer than five observations. A two-sided significance level of 0.05 was set.

## 5. Conclusions

We have developed and externally validated broadly applicable prediction models that accurately estimate the likelihood of beta-lactam target non-attainment in ICU patients within the first 12–36 h of therapy. These models are based on four readily available patient variables (age, sex, serum creatinine, and type of antibiotic) and help optimize the treatment of ICU patients with severe infections and can promote the efficient use of limited hospital resources. The RF and LR models showed the best performance among the three models tested. User-friendly online versions of the models have been developed to facilitate their bedside application (https://cator010.nl/betalactampredictor, accessed on 26 September 2023).

## Figures and Tables

**Figure 1 antibiotics-12-01674-f001:**
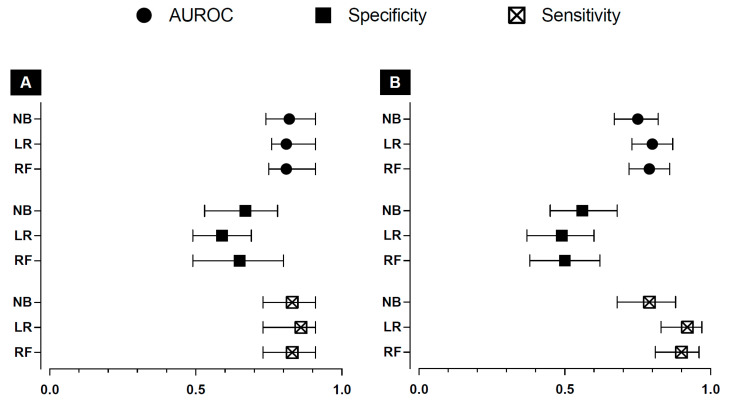
Performance metrics for the target non-attainment models in the internal validation data set (**A**) and the external validation data set (**B**) for the 0.20 threshold with 95% confidence intervals. NB: naïve Bayes; LR: logistic regression; RF: random forest; AUROC: area under the receiver operator curve.

**Figure 2 antibiotics-12-01674-f002:**
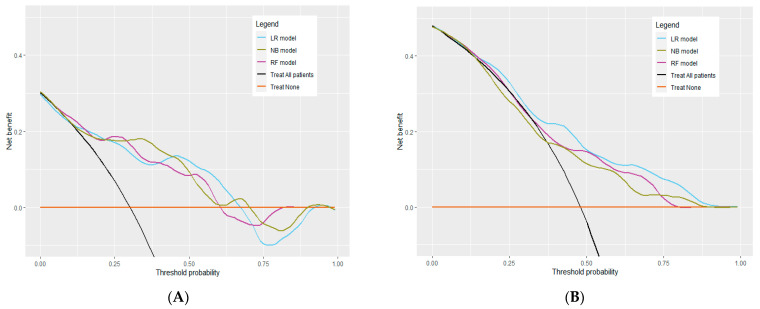
Decision curves for the target non-attainment models in the internal validation set (**A**) and the external validation set (**B**). Black line: “Treat all patients”. Brown line: “Treat none”. Blue line: net benefit of the logistic regression (LR) model. Green line: net benefit of the naïve Bayes (NB) model. Purple line: net benefit of the random forest (RF) model.

**Table 1 antibiotics-12-01674-t001:** Descriptive statistics of the patients used for model development and internal validation.

Variables	Overall (N = 376)
Age (years)	64.0 [56.0, 71.0]
Height (cm)	175 [167, 182]
Actual body weight (kg)	80.0 [69.9, 90.0]
Sex (male)	232 (61.7)
BMI (kg/m^2^)	25.8 [22.9, 29.3]
SOFA score d0	8.00 [5.00, 11.0] (N = 375)
APACHE IV score	50.0 [24.0, 78.0] (N = 375)
Cardiac surgery (No)	367 (97.6) (N = 370)
Trauma (No)	344 (91.5) (N = 370)
Study antibiotic	
Amoxicillin	15 (4.0)
Cefotaxime	77 (20.5)
Ceftazidime	15 (4.0)
Ceftriaxone	112 (29.8)
Cefuroxime	72 (19.1)
Flucloxacillin	17 (4.5)
Meropenem	54 (14.4)
Piperacillin–tazobactam	14 (3.7)
Serum creatinine d0 (mg/dL)	0.995 [0.690, 1.56]
Urea d0 (mmol/L)	9.00 [6.40, 14.7] (N = 365)
CRP d0 (mg/L)	136 [44.5, 251] (N = 367)
$\mathrm{WBC}\;d0\;(\times 10^{9}$/L)	13.2 [8.40, 17.9] (N = 373)
Albumin (g/L)	28.0 [22.0, 33.0] (N = 346)
The number of ICU days before start of the antibiotic	2.00 [2.00, 4.00]
Fluid balance d0	1240 [139, 2780] (N = 372)
Defined daily dosage	1.00 [1.00, 1.50] (N = 375)
Normalized dosing	1.00 [1.00, 1.50]

Values are presented as number (%) of patients or the median (IQR). Only when data are missing is the number of patients included in the analysis mentioned (N = X). Body mass index (BMI), Sequential Organ Failure Assessment score (SOFA), Acute Physiology and Chronic Health Evaluation score (APACHE IV), day of starting antibiotic therapy in the ICU (d0), white blood cells (WBCs), C-reactive protein (CRP), intensive care unit (ICU), the assumed average maintenance dose per day for a drug used for its main indication in adults (defined daily dosage), and dose of antibiotic divided by the defined daily dosage (normalized dosing).

**Table 2 antibiotics-12-01674-t002:** Odds ratios for the relevant variables on beta-lactam target non-attainment with the logistic regression model.

Variables	Odds Ratios	CI	*p* Value
(Intercept)	3.16	0.78–12.95	0.106
Serum creatinine	0.24	0.12–0.42	<0.001
Age	0.96	0.94–0.98	<0.001
Sex (male)	2.21	1.18–4.23	0.015
Amoxicillin	12.58	2.44–74.70	0.003
Cefotaxime	13.84	5.49–38.35	<0.001
Ceftazidime	0.47	0.02–3.22	0.510
Cefuroxime	6.25	2.47–16.97	<0.001
Flucloxacillin	10.86	2.84–46.90	0.001
Meropenem	2.64	0.92–7.69	0.072
Piperacillin–tazobactam	13.20	3.04–60.38	0.001

95% confidence interval (CI). For the type of antibiotic, ceftriaxone, as the most prescribed beta-lactam antibiotic, was used as reference.

## Data Availability

Data available on request due to privacy restrictions.

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
