# Peer review of "Predicting Beta-Lactam Target Non-Attainment in ICU Patients at Treatment Initiation: Development and External Validation of Three Novel (Machine Learning) Models"

_antibiotics, 2023, doi:10.3390/antibiotics12121674_

Round 1

Reviewer 1 Report

Comments and Suggestions for Authors

The article is well written and understandable from a scientific point of view. For this reason, I only have a few suggestions and comments that would improve the article. 

1) Why did the authors not include procalcitonin (PCT) in the analysis? Please explain.

2) It would be useful to also consider subgroups of the diseases (e.g. are there differences to oncological patients or patients with polytrauma). 

3) How many patients had sepsis or pneumonia? How many patients were ventilated? How many patients were given ß-lactams without severe signs of infection? 

4) Was it possible to establish a correlation with microbiological results (e.g. E. coli or St. aureus)? 

5) Would it also be useful to consider the known resistances in intensive care units? 

Reviewer 2 Report

Comments and Suggestions for Authors

1. Manuscript is well written and study is well conceptualised 

2.why hieght was taken as one of factor in model . How it affects Beta lactatms  physiology in body ? 

3. Liver function test and Kidney function ,Blood pressure and Cardiac functions may also have been used in analysis -Any such derangement in them may affect attainment of appropriate level of Beta lactams in body 

4. Cases with Other comorbidity (in ICU patients) like Tuberculosis , Diabetes may also be considered as factors 

5. Ethical clearance (document number must be included in script) 

6. What was criteria for choosing  specific 5-6 Beta lactam antibiotic only and leaving others (please justify) 

7 . does pk/Pd is a part of your models for evaluation? 

Reviewer 3 Report

Comments and Suggestions for Authors

Dear authors

Thnaks to let me review this manuscript.

The most important I do not have access to supp data.

From my part only a little details to conclude the manuscript

4.1.2. Pharmacodynamic target attainment --> could the author extend more about the LC-MS (model,paramaeter used, so on)

4.1.4. Model development --> it could great a short summary about each models (differences, limitations and advantages)

2.3. Online beta-lactam target non-attainment predictor --> if the author want to offer a kind webpage, my advice is to include a link for instructions. Sorry because I am mot sure what I shoud write .......

Linking to above mentioned, I miss a visual example during the discussion.

Thanks
